# Psychosocial functioning of adolescents with ADHD in the family, school and peer group: A scoping review protocol

**Kinga Karteczka-Świętek**[1]*, **Sylwia Opozda-Suder**[1], **Agnieszka Strojny**[2]

**1** Institute of Education, Faculty of Philosophy, Jagiellonian University in Kraków, Kraków, Poland, **2** Institute of Applied Psychology, Faculty of Management and Social Communication, Jagiellonian University in Kraków, Kraków, Poland

* kinga.karteczka@alumni.uj.edu.pl

## Abstract

### Objective

The objective of this scoping review is to investigate what is known about the psychosocial functioning of adolescents with ADHD. All basic life environments (family, school and peer group) will be considered.

### Introduction

Adolescence is especially critical for people with ADHD–natural maturation may be accompanied by changing symptoms of ADHD. A number of childhood difficulties transform during adolescence and contribute to problems in various areas that comprise psychosocial functioning. The available studies focus on selected domains of psychosocial functioning of adolescents with ADHD, however, to the authors' best knowledge, there is no comprehensive description of this issue. The lack of such a description is the main rationale for conducting this scoping review.

### Inclusion criteria

Pointing to the PCC elements (population, concept, context), the scoping review will include primary studies on the concept of psychosocial functioning (including functioning in the family, school and peer group). In the included studies, the term "psychosocial functioning" (or related) had to be used explicitly. The population will be adolescents (10 to 19 years old) with a formal diagnosis of ADHD (DSM classification) or Hyperkinetic Syndrome (ICD-9) or Hyperkinetic Disorders (ICD-10). There will be no restrictions on the research context.

### Methods

The methodology of scoping reviews will be applied in accordance with the guidelines of the Joanna Briggs Institute (JBI). The following databases: Academic Search Ultimate, ERIC, MEDLINE, ProQuest Central, PsycInfo, Scopus, and databases under the Web of Science will be searched for primary studies in peer-reviewed journals, written in English and

**Data Availability Statement:** All relevant data are within the paper and its Supporting Information files. Any additional information can be found on

the OSF project page (https://doi.org/10.17605/OSF.IO/6WX5C).

**Funding:** Author SOS received funding for a publication in Open Access in a competition procedure. The publication was funded by the Priority Research Area Society of the Future under the program "Excellence Initiative – Research University" at the Jagiellonian University in Krakow (www.futuresoc.id.uj.edu.pl) (FS.1.2.2021.101). The funders had no role in the study design, data collection and analysis, decision to publish, or preparation of the manuscript.

**Competing interests:** The authors have declared that no competing interests exist.

published since 1987. The analyses will be based mainly on frequency counts of the components of psychosocial functioning and population characteristics. The results will be presented in tabular form and supplemented with a descriptive summary. The protocol has been registered on the Open Science Framework: https://doi.org/10.17605/OSF.IO/MS82H [registration DOI].

## Introduction

At the very beginning of diagnosing what is known today as ADHD, it was considered a childhood disorder [1, 2]. Therefore, the vast majority of research on it has focused on children. Moreover, one of the first reviews [3] suggested a 50% decrease in the prevalence of ADHD every 5 years. This has led to the widespread belief that most people outgrow ADHD by young adulthood [4, 5]. The early findings are not reflected in subsequent reports. More recent studies have found that 60–85% of children diagnosed with ADHD continue to show symptoms during adolescence [6]. Based on the results of one of the latest systematic reviews [7], the prevalence of persistent ADHD in adulthood reported in studies published after 2011 is 55%. Thus, ADHD in many cases does not remit. These findings indicate the need to pay attention to the situation of adolescents with ADHD.

During adolescence, the process of sexual maturation and intense emotional and social development take place. This is a time to free oneself from parental influence, determine one's own identity and place within a peer group [8]. Young people with ADHD, like all teenagers, experience such changes. However, their natural development takes place in the presence of symptoms of the disorder. Additionally, this period of life may also involve substantial changes in the symptoms of the disorder [9–11]. Compared to childhood, in adolescence, signs of hyperactivity are less common and may be reported as feelings of restlessness, jitteriness, or impatience [12–15]. Research indicates that an increase in symptoms, including difficulties with attention, may be the consequence of the changes in the hormonal milieu during puberty [16]. Adolescents with ADHD are also more likely than their peers without ADHD to develop comorbidities such as conduct disorder or substance use disorders [12]. Additionally, anxiety disorders are common among teenagers suffering from ADHD [17]. The combination of ADHD and anxiety is associated with a very high risk of bipolar disorder in adulthood [18].

Emotional dysregulation and low frustration tolerance are also common in ADHD [13]. Adolescents with ADHD, compared to their peers, have significant difficulties coping with positive and negative emotions. They display stronger reactions to frustration or stress, emotional impulsivity, and rapid changes between emotions [19–21]. Poor emotion regulation may contribute to many negative outcomes, such as engagement in risky behaviour, including substance use, verbal and physical aggression, traffic accidents, and peer or family conflicts [19, 22, 23]. Adolescents with ADHD take more risk in daily life than their peers without ADHD [24]. This observation is supported by evidence on laboratory risk-taking tasks [25].

Therefore, childhood ADHD is associated with negative developmental outcomes in adolescence and adulthood [26] in various areas of life [27]. The studies revealed that children with ADHD have lower self-esteem than their peers [28] and are impaired within the peer functioning domain [29]. They are often unpopular with other children and may become isolated [30]. About 80% of them experience high rates of peer rejection and tend to have fewer friends [31, 32]. It should be noted that social problems, including lower peer acceptance, are associated with depressive symptoms in late childhood and adolescence [33, 34]. Moreover,

girls with ADHD have a higher rate of mood disorders than girls without ADHD [35]. In the school environment, children with ADHD experience greater difficulties than their typically developing peers [36], including academic problems [27, 35, 37], higher dropout rates or low academic achievement [38]. The relationship between a parent and an adolescent diagnosed with ADHD may also be impaired [39, 40]. Parents and teenagers with ADHD reported more conflict problems in their family, higher levels of anger intensity during conflicts, and more aggressive conflict tactics compared to a community control group [41]. Additionally, parent-child problems significantly mediate the relationship between attention problems and depression [42].

Summarising, the above description shows that teenagers with a childhood history of ADHD may encounter complex problems. These difficulties may have a negative impact on their psychosocial functioning. Hence, the concept of this scoping review is psychosocial functioning. This concept is considered an important construct in the conceptualisation of mental disorders. Accordingly, in the process of diagnosing ADHD, there must be clear evidence that the symptoms of the disorder have a direct negative impact on social, academic, or occupational functioning [12, 13].

There is no concrete definition of psychosocial functioning in the theoretical literature [43]. This concept usually refers to an individual's ability to achieve developmental tasks [44], interact with others and with society, and perform daily tasks [45]. The definition also takes into account an individual's satisfaction with this ability and their social interactions [45, 46]. With regard to children, their psychosocial functioning consists of their psychological development and interaction with the social environment [47]. Psychosocial functioning can also be narrowly defined by a specific domain (e.g. marital, school or occupational functioning) [45] or equate to social functioning [48] and described as the ability of an individual to function in various social roles, such as a student, family member, friend [46].

According to the literature on the subject [47, 49–55], researchers focus on different aspects of psychosocial functioning, including emotional health, self-esteem, social competences, social network, social adjustment, alienation, support, coping skills. By focusing on the particular life environments of adolescents, other components can be mentioned such as: family communication, child-parent relationship, learning motivation, academic performance, peer problems, peer rejection. Thus, the term psychosocial functioning is a very complex concept that includes a variety of constructs and covers a wide spectrum of variables. However, in order not to limit the scope of the review, this study does not adopt any definition of psychosocial functioning. The only definition-related requirement for this concept is that the authors of primary research articles had to use the term psychosocial functioning (or functioning in a family, school or peer group) explicitly. Further details are explained in the Concept subsection.

The first step in this scoping review was to verify the innovative character of the project. A preliminary search for ongoing or existing scoping reviews and systematic reviews on the psychosocial functioning of adolescents with ADHD has been conducted. At the end of August 2021, the following databases were searched: Cochrane Database of Systematic Reviews (CDSF), The Campbell Collaboration Database, The International Prospective Register of Systematic Reviews (PROSPERO), The Joanna Briggs Institute (JBI) EBP Database of Systematic Reviews and Implementation Reports, PubMed, The Open Science Framework (OSF) and figshare.com. The exemplary search strings for the PubMed and CDSF are presented in the S1 Appendix.

The preliminary searches have shown that still relatively little is known about the psychosocial functioning of adolescents with ADHD [56]. Among systematic or scoping reviews that in some way address the issue of psychosocial functioning in the ADHD population, almost all of

them combined children and adolescents into one group. Only two systematic reviews concerned adolescents with ADHD only, and both focused on assessing the effectiveness of interventions. One of the studies [52] was related to the peer functioning. The other [57] focused on the academic domain and limited the research design to randomised controlled trials. Therefore, both reviews were limited to selected areas of psychosocial functioning, which narrowed the spectrum of this concept. Moreover, none of them was intended to describe psychosocial functioning comprehensively, as is the case in the proposed review.

In conclusion, it is rational to conduct this scoping review. According to the preliminary search and to the best knowledge of the authors, this is the first scoping review exploring the concept of psychosocial functioning of adolescents with ADHD in three basic life environments simultaneously. This holistic description argues for an innovative character of the present project. Another argument for the innovative nature of the project is that it focuses on adolescence. The facts presented previously clearly show that the situation of adolescents with ADHD is difficult and complicated. The increased risk of negative effects in all basic life environments is reflected in the psychosocial functioning of teenagers with ADHD. As mentioned, the importance of assessing psychosocial functioning in ADHD is acknowledged in the DSM and ICD classification systems. Socialization, understood as a process of continuous interactions between a maturing person and their family, school and peer environment, stimulates the development, adaptation and functioning of the individual. Deterioration in psychosocial functioning has adverse consequences for the development of an individual in various areas, including cognition, body image, mental and physical health [58–60]. Therefore, it seems justified to broadly explore the issue of psychosocial functioning of adolescents with ADHD. The scoping review is one of the newest research approaches that makes it possible in a structured manner.

## Objectives of the study

The main goal of the scoping review is to describe comprehensively the concept of psychosocial functioning of adolescents with ADHD. This description will contain information both on psychosocial functioning in general and in particular life environments (family, school, peer group). Gathering the information will help to organise the existing evidence and identify gaps in the knowledge of the concept. Evidence-based detailed knowledge collected is needed as a starting point for effective profiling of educational and therapeutic activities.

The first stage of achieving the main goal will be to find the main components and methods of measuring the concept of "psychosocial functioning of adolescents with ADHD". The results of the scoping review will show how researchers operationalise the concept, what aspects they focus on, and what indicators dominate.

Additionally, conducting the scoping review will help to verify the feasibility of systematic reviews (and possibly meta-analyses): (1) on the effects of non-medical interventions on psychosocial functioning of adolescents with ADHD and (2) on potential differences in psychosocial functioning from their peers without ADHD.

### Review questions

This scoping review will answer the following primary question: What is known about the psychosocial functioning of adolescents with ADHD? The research sub-questions are as follows:

1. What are the constitutive components of the psychosocial functioning of adolescents with ADHD? (In other words: How is "psychosocial functioning of adolescents with ADHD" operationalised?)

2. What characterises the psychosocial functioning of adolescents with ADHD in general?

3. What characterises the psychosocial functioning of adolescents with ADHD in the family?

4. What characterises the psychosocial functioning of adolescents with ADHD at school?

5. What characterises the psychosocial functioning of adolescents with ADHD in the peer group?

## Materials and methods

The proposed scoping review will be conducted in accordance with the Joanna Briggs Institute (JBI) methodology for scoping reviews [61]. All data will be presented in accordance with The Preferred Reporting Items for Systematic Reviews and Meta-Analyses extension for Scoping Reviews (PRISMA-ScR) [62]. The PRISMA-ScR checklist is attached as S2 Appendix. The protocol has been registered on the Open Science Framework: https://doi.org/10.17605/OSF.IO/MS82H [registration DOI]. The scoping review will be based only on the results already published in peer-reviewed articles. Therefore, no primary data will be collected, and ethical approval was not required.

### Eligibility criteria

The studies included in the scoping review must meet all of the criteria described below.

**Participants.** The population covered by this scoping review is adolescents with a formal diagnosis of Attention-Deficit/Hyperactivity Disorder (ADHD; based on DSM criteria) or Hyperkinetic Syndrome (HKS; based on ICD-9 criteria) or Hyperkinetic Disorders (HKD; based on ICD-10 criteria).

Research will be included regardless of participants' completed or ongoing therapies, ADHD subtypes (presentations), comorbidities or medications. However, all these circumstances will be noted. Studies that focused on a different disorder (e.g. autism spectrum disorders, oppositional defiant disorder, conduct disorder), but included participants with comorbid ADHD, will only be eligible for inclusion if results were presented for the subgroup diagnosed with ADHD.

According to definitions of adolescence of the World Health Organization (WHO) and the American Psychological Association (APA), an age range of 10 to 19 will be used. Studies where the age of the participants exceeds this range will only be included if the results have been reported separately for age groups. Therefore, it will be possible to extract the data for the group of interest (aged from 10 to 19).

**Concept.** The scoping review will focus on psychosocial functioning of adolescents with ADHD, taking into account the three most important life environments–family, school and peer group.

Studies will be included if psychosocial functioning has been measured and reported results. The only definition-related inclusion criterion for the concept is that the authors of the primary studies had to explicitly use one of the following terms for the variables of interest: psychosocial functioning, social functioning, family functioning, school functioning, academic functioning, peer functioning, peer group functioning, or adequate grammatical forms. All the considered phrases were included in the search strings (see S3 Appendix). It should be emphasized that, in order not to narrow the scope, this review does not adopt a specific definition of psychosocial functioning. The authors of the review are aware that psychosocial functioning is a very complex concept with no concrete semantic boundaries. Consequently, they will not decide on the definition of the concept and will focus on finding the main components and methods of measuring psychosocial functioning in the included studies.

**Context.** There will be no restrictions regarding the context of the studies. All geographic locations, cultural/sub-cultural, sociodemographic or economic contexts will be included.

**Types of sources.** All study designs will be eligible for the scoping review, including self-report data as well as data obtained from parents, teachers, therapists. However, searches will be limited to full text articles on primary research in peer-reviewed journals. Therefore, systematic reviews and meta-analyses will be excluded, as well as commentaries, posters, opinion papers or "grey literature".

## Exclusion criteria

The basis for excluding studies from the scoping review will be if any of the following criteria are met:

1. only informal diagnosis of ADHD (e.g. from a parent's or teacher's report or any ADHD rating scale other than the DSM or ICD criteria);

2. only participants under 10 or over 19 years old, or no possibility for extracting data for the age group of interest;

3. no possibility to extract data only for the ADHD subgroup (in studies with comorbid ADHD);

4. no use of any of the following terms for the variables of interest: psychosocial functioning, social functioning, family functioning, school functioning, academic functioning, peer functioning, peer group functioning, or adequate grammatical forms;

5. paper in a language other than English (see the Search strategy section for details);

6. non-peer reviewed journal;

7. research published before 1987 (see the Search strategy section for details);

8. no possibility to access the full version of the text or the data being searched.

## Search strategy

The search strategy will aim to locate only primary studies in peer-reviewed journals, written in English and published since 1987.

▶ In order to ensure methodological correctness and high quality of the researches, it was assumed that only peer-reviewed articles would be taken into account. Therefore, the search for "grey literature" will not be carried out.

▶ Systematic reviews and meta-analyses will be excluded for practical reasons in order to limit the volume of material to be retrieved and reviewed. Moreover, the inclusion of this type of articles would likely give very similar (or even identical) results to an electronic search limited to primary studies only.

▶ Due to the linguistic capabilities of the team and common practice, only English language papers will be considered.

▶ The name ADHD has been used since 1987, when the DSM-III-R was published. Additionally, it was noticed at that time that the disorder does not pass with childhood. This change in approach was reflected in the book with a meaningful title: "*The hyperactive child*, *adolescent*, *and adult*: *Attention deficit disorder through the lifespan*" [63]. Accordingly, publishing since 1987 is a limitation for the scoping review.

An initial limited search of MEDLINE and ERIC was undertaken to identify articles about adolescents with ADHD and/or psychosocial functioning. The text words contained in the titles and abstracts of relevant articles, and the index terms used to describe these papers, all combined on the basis of Boolean logic, were used to develop a full search strategy for databases under EBSCO and Scopus (included in the S3 Appendix).

The titles, abstracts and keywords will be searched in the following databases: Academic Search Ultimate (via EBSCO), ERIC (via EBSCO), MEDLINE (via EBSCO), ProQuest Central, PsycInfo, Scopus, and all databases under the Web of Science (excluding MEDLINE).

The search strategy, including all identified keywords and index terms, will be adapted for each included database. The reference list of all sources of evidence included in the review will be screened for additional studies. A re-run search will also be conducted prior to final analyses to capture the latest articles.

## Source of evidence selection

Following the search, all identified citations will be collated and uploaded into Zotero 5.0 [64] and duplicates will be removed. Pilot tests will be conducted to increase consistency among reviewers. In the first one, 10% of all included citations (specifically, titles and abstracts) will be independently screened by all three reviewers against the review inclusion criteria. They will then discuss the results, amend the screening and modify the eligibility criteria as necessary. After the first pilot test, the titles and abstracts of the remaining 90% of citations included will be independently screened by two reviewers against the inclusion criteria. Potentially relevant sources will be retrieved in full and their citation details imported into the new collection in Zotero.

The second pilot test will cover 10% of the full text articles. All three reviewers will independently screen this portion of the resources to determine if the inclusion criteria are met. Then, the remaining 90% of the full texts of the selected citations will be assessed in detail by two independent reviewers against the inclusion criteria. Reasons for the exclusion of the sources of evidence in full text that do not meet the inclusion criteria will be recorded and reported in the scoping review. Any doubts or disagreements between reviewers at any stage of the selection process will be resolved through discussion and consensus of the research team.

The results of the search and the study inclusion process will be reported in full in the final scoping review and presented in a Preferred Reporting Items for Systematic Reviews and Meta-Analyses Extension for Scoping Review (PRISMA-ScR) flow diagram [62], supplemented with the number of sources collected for each review sub-question.

## Data extraction

Data will be extracted from papers finally included in the scoping review by three reviewers using their data extraction tool (Excel spreadsheet). The data extracted will include specific details about the participants, concept, context, study methods and key findings relevant to the review questions.

The pilot of the draft extraction form will be carried out on a randomly selected 10% of the articles included in the review. All three reviewers will independently extract data from this portion of the resources. In case of any doubts, the whole team will make a decision through discussion and the data extraction form will be improved. Then the remaining 90% of the articles will be divided among the three reviewers and each of them will extract the data from their part.

A draft extraction form is provided (see S4 Appendix). The draft data extraction tool will be modified and revised as necessary during the process of extracting data from each included

evidence source. Modifications will be detailed in the scoping review. If appropriate, authors of papers will be contacted to obtain missing or additional data, where required.

## Data analysis and presentation

First, the extracted results will be divided into overall psychosocial functioning and functioning in particular life environments (family, school and peer group). The sorted research evidence will be summarised and descriptively mapped to characterise components of psychosocial functioning. This will be based on the occurrence of the variables measured. The following frequency counts will also be used: components of psychosocial functioning, variables or measurement tools assessing psychosocial functioning, available population characteristics (including country, age, gender, adolescence stage, ADHD presentation/subtype, comorbidities), pharmacological or other treatments/interventions, study designs, or other important data fields (unknown at protocol stage).

Summarising the research evidence on the topic, an attempt will be made to broadly describe psychosocial functioning of adolescents with ADHD in general. However, this will not be an in-depth type of analysis because this would be beyond the scope of the scoping review. Analyses will be based on examining the similarities and differences between the research evidence, the occurrence of variables or measurement tools commonly used to assess psychosocial functioning, the identification of research gaps and any key findings that would be useful to answer the question of what is known about the scoping review topic.

The data will be presented in tabular form and, if necessary, graphically. A preliminary tabular presentation of individual PCC elements is included in S5 Appendix. The tabular mapping technique will be used to present how psychosocial functioning was operationalised in included evidence sources (see S6 Appendix). A narrative summary will accompany the tabulated or charted results and will describe how the results relate to the review objectives and questions.

Overlaps and/or gaps in the literature will be identified. Implications for future studies will be suggested, mainly based on research gaps. A discussion on the need and feasibility of future research in the context of psychosocial functioning, in particular systematic reviews (and possibly meta-analyses) of the treatment effectiveness in adolescents with ADHD and potential differences from peers without ADHD, will also be carried out.

## Discussion

The protocol presented in this paper describes a detailed plan for conducting the scoping review on psychosocial functioning of adolescents with ADHD. Publishing the protocol at the beginning of the study ensures that information about it reaches a wide range of researchers, and will enable the reproduction of this scoping review in the future.

Necessary changes to the study protocol will be presented in the final scoping review report, which is planned to be published in a peer-reviewed journal. The article will include full results of the search: a description of the study inclusion and data extraction process as well as the findings for scoping review questions. At this time all available data will be made public. It should also be mentioned that any disagreements or uncertainties that will arise between the team members at any stage of the research process will be resolved through discussion. As needed, amendments to the study will be updated on the OSF project page https://doi.org/10. 17605/OSF.IO/6WX5C [page DOI].

Pointing to the strengths of the study, applying a scoping review methodology, the latest research approach, enables the study of a wide spectrum of psychosocial functioning. Thus, the family, school and peer environment are taken into account at the same time, which gives

a comprehensive description. Moreover, focusing on adolescence attests to the innovative character of the study. Additionally, the time span of the search is very wide (since 1987) and there are no restrictions on the context of the research included. It is also worth emphasizing that conducting the scoping review will help to verify the feasibility of (future) systematic reviews (and possibly meta-analysis): (1) on the effects of non-medical interventions on psychosocial functioning of adolescents with ADHD and (2) on potential differences in psychosocial functioning from their peers without ADHD. These reviews will contribute to an even deeper knowledge of the subject.

However, the authors are aware that the planned study is not free of limitations. One of the issues that may be debatable is including only participants with a formal diagnosis of ADHD. This will result in the rejection of some studies, but the authors are convinced that this approach will contribute to collecting the most homogeneous sample possible. Another potentially questionable decision is not to search "grey literature". The argument for such a solution was the desire to include only highly reliable, well-designed research. Moreover, by balancing the breadth of the review with the reasonable time frame and research project resources, unpublished studies were excluded. A decision like this was also made in other scoping reviews [65–69]. The exclusion of systematic reviews and meta-analyses in this scoping review may also be considered a limitation. A search including this type of articles would likely give corresponding (or even identical) datasets of primary studies to the search excluding systematic reviews and meta-analyses–assuming, of course, that both searches would have a similar strategy and inclusion criteria. If the search strategies and inclusion criteria were different–it is possible that dataset from the search including systematic reviews and meta-analyses would contain additional primary studies. However, these differences in search approaches would make the scoping review procedure less precise and less replicable. And the authors wanted to avoid it. There are also other scoping reviews that exclude systematic reviews and meta-analyses, focusing searches solely on primary studies [65–67, 70]. The review may also be limited by excluding non-English articles. The language criterion was based on the capabilities of the team members. Moreover, it is a common practice in reviews.

Despite the indicated limitations, the proposed scoping review, to the authors' best knowledge, will be the first to investigate the concept of psychosocial functioning of adolescents with ADHD simultaneously in three basic life environments, assuming precise methodological criteria.

## Supporting information

**S1 Appendix. The exemplary search strings for a preliminary search.**
(PDF)

**S2 Appendix. The PRISMA-ScR checklist.**
(PDF)

**S3 Appendix. Search strings for Scopus and EBSCO databases.**
(PDF)

**S4 Appendix. Data extraction instrument (preliminary table).**
(PDF)

**S5 Appendix. Draft tabular presentation of the PCC components.**
(PDF)

**S6 Appendix. Components of psychosocial functioning in evidence sources (draft table).**
(PDF)

## Acknowledgments

The authors would like to thank Professor Franciszek Wojciechowski, irreplaceable doctoral supervisor, for his support and trust.

## Author Contributions

**Conceptualization:** Kinga Karteczka-Świętek.

**Investigation:** Kinga Karteczka-Świętek.

**Methodology:** Kinga Karteczka-Świętek, Sylwia Opozda-Suder.

**Project administration:** Kinga Karteczka-Świętek.

**Writing – original draft:** Kinga Karteczka-Świętek.

**Writing – review & editing:** Kinga Karteczka-Świętek, Sylwia Opozda-Suder, Agnieszka Strojny.

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
