## [Decision Letter · Decision Letter 0]

22 Mar 2022

PONE-D-21-37767Psychosocial functioning of adolescents with ADHD in the family, school and peer group: A scoping review protocolPLOS ONE

Dear Dr Karteczka-Swietek,

Thank you for submitting your manuscript to PLOS ONE. After careful consideration, we feel that it has merit but does not fully meet PLOS ONE’s publication criteria as it currently stands. Therefore, we invite you to submit a revised version of the manuscript that addresses the points raised during the review process.

Please be sure to address all of the comments made by the reviewers. 

Please submit your revised manuscript by May 05 2022 11:59PM.  If you will need more time than this to complete your revisions, please reply to this message or contact the journal office at plosone@plos.org. Please include the following items when submitting your revised manuscript:A rebuttal letter that responds to each point raised by the academic editor and reviewer(s). You should upload this letter as a separate file labeled 'Response to Reviewers'.A marked-up copy of your manuscript that highlights changes made to the original version. You should upload this as a separate file labeled 'Revised Manuscript with Track Changes'.An unmarked version of your revised paper without tracked changes. You should upload this as a separate file labeled 'Manuscript'.

We look forward to receiving your revised manuscript.

Kind regards,

Gerard Hutchinson, MD

Academic Editor

PLOS ONE

Journal Requirements:

Reviewers' comments:

Reviewer's Responses to Questions

**Comments to the Author**

1. Does the manuscript provide a valid rationale for the proposed study, with clearly identified and justified research questions?

Reviewer #1: Partly

Reviewer #2: Yes

2. Is the protocol technically sound and planned in a manner that will lead to a meaningful outcome and allow testing the stated hypotheses?

Reviewer #1: Partly

Reviewer #2: Yes

3. Is the methodology feasible and described in sufficient detail to allow the work to be replicable?

Reviewer #1: Yes

Reviewer #2: Yes

4. Have the authors described where all data underlying the findings will be made available when the study is complete?

Reviewer #1: Yes

Reviewer #2: Yes

5. Is the manuscript presented in an intelligible fashion and written in standard English?

Reviewer #1: No

Reviewer #2: Yes

6. Review Comments to the Author

You may also provide optional suggestions and comments to authors that they might find helpful in planning their study.

Reviewer #1: Thank you for the opportunity to review this manuscript of a scoping review protocol. I would recommend this manuscript undergo some major revisions in order to be published. My major areas of concern are:

1. Some sentences/paragraphs are difficult to understand or quite vague, which I assume are language issues. It would be good to get a native speaker to review the manuscript if possible and help to improve it

2. I’m not entirely sure what the rationale is for this scoping review and what the authors are hypothesising – this may be a language issue but it’s really important that the authors make it clearer why they are conducting this review and what they want to find out.

3. There is not definition of the main concept “Psychosocial functioning”. The manuscript centres around this concept but it is unclear to me how exactly this is conceptualised for this review.

4. The introduction reads very vague on a lot of aspects. There are quite a few blanket statements made without evidence which could, potentially be quite harmful for children and adolescents with ADHD. I have highlighted a few issues further down.

Some more detailed comments below:

Abstract

P1,l20: hyperkinetic disorder is commonly differentiated from ADHD. I would suggest sticking with one of the terms. I would recommend ADHD, as hyperkinetic disorder is, to my knowledge, a bit outdated.

P1, l20-23, Introduction: seems a bit vague. It would be good to have a definition of “psychosocial functioning” in there.

P2,l25-28: in the text you mention that you will only include studies in English and from 1987 onwards but here you state that all primary studies are included. The abstract should align with the rest of the manuscript.

P2, l27/28. “The concept is derived from the social sciences.” – which concept and why is this information important here?

P2, l28: “There will be no restrictions on the research context” – again, in the text you mention certain exclusions (medical, neurobiological, neurocognitive) which don’t seem to be reflected here.

p2, l31-32: no need for brackets around the databases, the way it’s phrased they should be listed after a colon.

P2,l34: “frequency counts” of what though? Can you explain the method of analysis a bit better?

Introduction

I could not find your review question and/or aim stated explicitly in the introduction section. It would be very helpful to include a research question or hypothesis in the introduction. What is it that you are actually wanting to find out and why?

P2, l43/44: Consider changing this sentence to “may involve changes in the symptoms”, it is very general and may not be the case for everyone. I recommend deleting the next sentence (“Therefore, all natural processes are affected…”) as this is not correct and no evidence is provided for this.

P2/3,l46/47: citation needed.

P2/3,l47/48: “ADHD does not pass in most cases…” This would be highly contested. There are various studies that show a large percentage of children who outgrow their ADHD-related behaviours throughout adolescence. A proportion of children struggle with ADHD behaviours throughout adulthood, but this is not the case for many. Also, ADHD is defined by the displayed symptoms/behaviours, so if those are no longer present, the person has, by definition, no ADHD. Thus, I suggest amending this statement.

P2,l48-50: “Moreover, children with hyperkinetic disorder have limited contact with their peers, isolate themselves and due to their behaviour or the problems they cause, they have no close friends outside the family.” This sentence needs deleting or a complete rewrite. 1. Hyperkinetic disorder is no longer a diagnosis used and I would recommend only referring to it in your inclusion criteria to capture older studies that use ICD-10 or before definitions. Commonly, hyperkinetic disorder is often distinguished as a more severe form of ADHD, I think it may lead to confusion, especially amongst non-European readers, if you switch between the two terms throughout the text. 2. I think it is very dangerous (and wrong) to state that children (or anyone else) with ADHD have no friends and have limited contact because of the behaviours and that they cause problems. Please think about what you are implying with this. There are many children with ADHD that lead pretty happy lives and are socially completely integrated.

P3,l53: “The above facts clearly show…” I don’t think you have shown this above. Please consider re-writing your introduction to detail how ADHD impacts adolescents.

P3,l65-71: This whole paragraph should be moved into the methods section, it is not relevant to the introduction. Consider including more information on the actual concept of “psychosocial functioning” – a definition of how you utilised this concept for this study would be very helpful. It is not quite clear to me what you mean by this exactly.

P4/5,l80-101: I would suggest shortening this a bit and summarising more concisely what has been done and what has not been done. It is not quite clear to me what the new aspect of your work is for this. It reads as if there has already been a lot of studies and reviews of psychosocial functioning done. I would contest your last sentence in this paragraph: there is a lot of research out there on ADHD and we know a lot about it already. Arguably, what we still do not know (or what is still contested) is what sort of interventions help adolescents with ADHD.

Objectives of the study and review questions

Consider moving both of these into the introduction. I was looking for this information earlier on. According to JBI these should be in the introduction of a protocol.

P5,l105/6: “this study assumes taking into account..” this sentence doesn’t really make sense – I would recommend deleting it.

The objectives section still does not make very clear what the actual main aim of this study is apart from gathering information on psychosocial functioning in adolescents with ADHD. Why do you want to gather this information?

Eligibility criteria

Concept, p7,l158: There really needs to be some explanation of “psychosocial functioning” – this is your main concept. There needs to be a definition (see also line163 in which you say “as defined in this review” but you haven’t defined it).

The mentioning of the three environments could be moved to the context section – this is the context in which you are looking at the concept.

P7, l160-166: This is all not relevant to the concept. This can go into exclusion criteria.

P8, l175/176: why are you excluding systematic reviews and meta-analyses?

P8,l188/193: for these exclusion criteria it is, again, really important that you have a solid definition of psychosocial functioning somewhere. Just presenting some examples and then writing “etc”., does not seem a full list of exclusions or inclusion. Also, if you are excluding all studies without clinical diagnosis and also those that do not fully conceptualise psychosocial functioning in a “holistic” way (however you want to define that) -my guess is you will be left with very few or no studies.

Source of evidence selection

P10 – according to JBI methodology the title/abstract screening and full text selection should be done by 2 reviewers independently.

Data analysis and presentation

P12,l 272: “Although the synthesis of the results is not the goal of the scoping review,..”: what is your goal then? You may need to re-phrase this. The goal of any review should be the synthesis of evidence in some form!

Reviewer #2: 1. Summary of the research and your overall impression

The authors describe a planned scoping review that explores an important area in the field of adolescent mental health and wellbeing. They clearly outline the methodological steps that will be undertaken to identify and synthesise studies that discuss the topic of psychosocial difficulties among adolescents with ADHD. Bringing together the literature on 3 key areas of adolescents’ life (i.e. family, school and peer group) would be a valuable contribution to knowledge that could inform future research. However, there are a few minor issues that could be addressed at this proposal stage which could benefit the study process and outcomes.

2. Minor issues to be considered

• The authors stated that other/similar review have been conducted “relatively long ago”. Would it be possible for the authors to explicitly state a timeframe? For example, “more than 5 years ago”.

• The authors used “etc” a number of times in the manuscript. Can the authors rephrase these sentences to avoid using “etc” as this is less common for academic submissions?

• The authors describe the propose study as innovative. However, the argument that “ADHD is still being researched and we still know relatively little about it in youth” is a bit unclear. The authors may wish to further develop this point possibly narrowing it to focus on psychosocial difficulties in adolescents. This is important as there is a wealth of evidence on ADHD and youth as it relates to other fields like medicine and nutrition.

• The authors may wish to provide a citation to support their justification for excluding information not published in peer reviewed journals. This is important as one of the advantages of the scoping review methodology is that is gives researchers the opportunities to search more broadly using grey literature sources to identify information that is usually missed in traditional systematic reviews. It is possible that grey literature searches were conducted in the preliminary searches as the authors mentioned Figshare and OSF.

• The authors described piloting 10% of the studies during screening and data extraction before independently working on the remaining 90%. However, it is not clear if any verification or cross-checking or comparisons will be done to ensure further accuracy, reliability and consistency. Although not explicitly stated in the JBI guidance would the authors consider this as a limitation?

• Similarly, the authors may wish to acknowledge that not including the “optional consultation phase” proposed by other scoping review methodologist (Arksey and O’Malley framework) could be a limitation or a consideration for future research? Owing to the research questions proposed there might be an opportunity here for valuable input from lived experience experts.

7. PLOS authors have the option to publish the peer review history of their article (what does this mean?). If published, this will include your full peer review and any attached files.

Reviewer #1: No

Reviewer #2: **Yes: **Shaun Liverpool

---

## [Author Response · Author response to Decision Letter 0]

5 May 2022

RESPONSE TO REVIEWERS

Dear Academic Editor and Reviewers,

We would like to thank you for the opportunity to revise our manuscript and for the helpful comments and constructive suggestions. After careful consideration, we have responded to each comment and heeded your feedback. It is our belief that the manuscript has gained in quality after making the edits suggested. 

The descriptions of the changes made are included below. This content is also available in a more accessible form in the attached file "Response to Reviewers.pdf".

The line numbering refers to the version of the text subjected to review (first submission).

ACADEMIC EDITOR:

Answer: We have checked and corrected the manuscript according to PLOS ONE's style requirements.

REVIEWER #1:

1. Some sentences/paragraphs are difficult to understand or quite vague, which I assume are language issues. It would be good to get a native speaker to review the manuscript if possible and help to improve it

Answer: Thank you for the suggestion, our paper has been checked and corrected by an experienced native British English proofreader.

2. I’m not entirely sure what the rationale is for this scoping review and what the authors are hypothesising – this may be a language issue but it’s really important that the authors make it clearer why they are conducting this review and what they want to find out.

Answer: Thank you for drawing attention to this issue. We have expanded the last paragraph before the Objectives of the study subsection to emphasize the rationale for conducting our scoping review. According to the JBI methodology, we do not pose any hypotheses, instead there are review questions. The Objectives of the study subsection was modified to describe in detail what we want to achieve.

We hope that the changes have clarified this issue.

3. There is not definition of the main concept “Psychosocial functioning”. The manuscript centres around this concept but it is unclear to me how exactly this is conceptualised for this review.

Answer: Thank you for pointing out this deficiency. We have changed two paragraphs in the Introduction to briefly outline possible ways of defining psychosocial functioning present in the literature. However, we also emphasized that:

“(…) in order not to limit the scope of the review, this study does not adopt any definition of psychosocial functioning. The only definition-related requirement for this concept is that the authors of primary research articles had to use the term psychosocial functioning (or functioning in a family, school or peer group) explicitly. Further details are explained in the Concept subsection.”

We have also expanded the Concept subsection to better explain our research assumptions. We hope that the changes made are sufficient.

4. The introduction reads very vague on a lot of aspects. There are quite a few blanket statements made without evidence which could, potentially be quite harmful for children and adolescents with ADHD. I have highlighted a few issues further down. Some more detailed comments below.

Answer: Thank you very much for drawing attention to this issue. In response to this comment, we have almost completely rewritten the Introduction and supported it with evidence based on research. Detailed answers to the highlighted issues are provided below. In our opinion, all the changes made have streamlined and cleared the Introduction.

4.1. Abstract

A. P1,l20: hyperkinetic disorder is commonly differentiated from ADHD. I would suggest sticking with one of the terms. I would recommend ADHD, as hyperkinetic disorder is, to my knowledge, a bit outdated.

Answer: Thank you for the suggestion. Wherever possible, “hyperkinetic disorders” has been replaced with “ADHD”. The term “hyperkinetic disorders” remained in the inclusion criteria. This is necessary because this term was the official name for ADHD in the ICD-10, and the ICD is a widely used classification system in many places around the world (including European countries, Australia, Canada, Brazil, China, Korea, South Africa). Only since January 1, 2022, when the ICD-11 came info effect, “ADHD” has replaced “hyperkinetic disorders”. Moreover, for precision, we have also added the term “hyperkinetic syndrome” (to reflect the ICD-9 classification in the inclusion criteria).

B. P1, l20-23, Introduction: seems a bit vague. It would be good to have a definition of “psychosocial functioning” in there.

Answer: We have made a few changes to make this paragraph more precise. Moreover, we have provided additional information on the concept in the Inclusion criteria paragraph. However, as required by PLOS ONE, the Abstract cannot exceed 300 words. Therefore, no more information could be added.

C. P2,l25-28: in the text you mention that you will only include studies in English and from 1987 onwards but here you state that all primary studies are included. The abstract should align with the rest of the manuscript.

Answer: Thank you for pointing out this lack of precision. This confusing information and the entire paragraph have been corrected. In addition, we have emphasized that (in this paragraph) there are only criteria related to the PCC elements (population, concept, context).

D. P2, l27/28. “The concept is derived from the social sciences.” – which concept and why is this information important here?

Answer: The “concept” was about psychosocial functioning. This information was related to the PCC elements (which should be listed in the Inclusion criteria paragraph). This sentence was just additional information. However, it was unnecessary and imprecise and has been removed.

E. P2, l28: “There will be no restrictions on the research context” – again, in the text you mention certain exclusions (medical, neurobiological, neurocognitive) which don’t seem to be reflected here.

Answer: This sentence is correct – there will be no restrictions on the research context. However, the issue of the context of the scoping review was indeed described unclear in our protocol. This problem has been corrected throughout the manuscript, mainly by removing sentences related to medical, neurobiological or neurocognitive studies from the Introduction, Concept and Exclusion criteria subsections.

F. p2, l31-32: no need for brackets around the databases, the way it’s phrased they should be listed after a colon.

Answer: Thank you for this comment. The suggested change has been made.

G. P2,l34: “frequency counts” of what though? Can you explain the method of analysis a bit better?

Answer: Thank you for drawing attention to this understatement. We have added some general information (in a way that the limited length of the Abstract allowed). A detailed description is provided in the Data analysis and presentation subsection of the manuscript.

4.2. Introduction

A. I could not find your review question and/or aim stated explicitly in the introduction section. It would be very helpful to include a research question or hypothesis in the introduction. What is it that you are actually wanting to find out and why?

Answer: The Objectives of the study and the Review questions are subsections of the Introduction section (which is in line with the JBI manual). Indeed, it is hard to remark this in the submitted manuscript file. According to PLOS ONE's style requirements, the main section (Introduction) has the headline Times New Roman 18 and the subsections have the headings Times New Roman 16. Additionally, the entire Objectives of the study subsection has been corrected (see Answer #4.3.C), and we hope that after these changes our goals are clearly defined.

B. P2, l43/44: Consider changing this sentence to “may involve changes in the symptoms”, it is very general and may not be the case for everyone. I recommend deleting the next sentence (“Therefore, all natural processes are affected…”) as this is not correct and no evidence is provided for this.

Answer: Thank you for this comment. As suggested, the first sentence has been changed (additionally, more citations were provided). The second sentence has not been deleted but replaced by the following:

“However, their natural development takes place in the presence of symptoms of the disorder.”

C. P2/3,l46/47: citation needed.

Answer: As a consequence of the changes made throughout the Introduction, this sentence has been rewritten, supported by citations and moved to the beginning of the Introduction.

D. P2/3,l47/48: “ADHD does not pass in most cases…” This would be highly contested. There are various studies that show a large percentage of children who outgrow their ADHD-related behaviours throughout adolescence. A proportion of children struggle with ADHD behaviours throughout adulthood, but this is not the case for many. Also, ADHD is defined by the displayed symptoms/behaviours, so if those are no longer present, the person has, by definition, no ADHD. Thus, I suggest amending this statement.

Answer: Thank you for focusing attention on this issue. This statement has been amended, expanded and supported by citations from research results. As a consequence of these changes, the entire paragraph has been created, which has been placed at the beginning of the Introduction.

E. P2,l48-50: “Moreover, children with hyperkinetic disorder have limited contact with their peers, isolate themselves and due to their behaviour or the problems they cause, they have no close friends outside the family.” This sentence needs deleting or a complete rewrite. 

1. Hyperkinetic disorder is no longer a diagnosis used and I would recommend only referring to it in your inclusion criteria to capture older studies that use ICD-10 or before definitions. Commonly, hyperkinetic disorder is often distinguished as a more severe form of ADHD, I think it may lead to confusion, especially amongst non-European readers, if you switch between the two terms throughout the text. 

2. I think it is very dangerous (and wrong) to state that children (or anyone else) with ADHD have no friends and have limited contact because of the behaviours and that they cause problems. Please think about what you are implying with this. There are many children with ADHD that lead pretty happy lives and are socially completely integrated.

Answer: 1. Thank you for the suggestion. The term “hyperkinetic disorders” has been replaced by “ADHD” (see Answer #4.1.A for details). 

2. Thank you for highlighting the careless wording that we have used which may actually be harmful. This sentence has been completely rewritten (using research results only) and supported by citations.

F. P3,l53: “The above facts clearly show…” I don’t think you have shown this above. Please consider re-writing your introduction to detail how ADHD impacts adolescents.

Answer: This part of the Introduction has also been rewritten and expanded to highlight how ADHD impacts adolescents. We hope that the changes made have improved this issue.

G. P3,l65-71: This whole paragraph should be moved into the methods section, it is not relevant to the introduction. Consider including more information on the actual concept of “psychosocial functioning” – a definition of how you utilised this concept for this study would be very helpful. It is not quite clear to me what you mean by this exactly.

Answer: Thank you for the comment. To clarify this part of the Introduction, this paragraph has been deleted. Moreover, we have added information about possible ways of defining psychosocial functioning present in the literature and that our study does not adopt any definition of psychosocial functioning (see Answer #3).

H. P4/5,l80-101: I would suggest shortening this a bit and summarising more concisely what has been done and what has not been done. It is not quite clear to me what the new aspect of your work is for this. It reads as if there has already been a lot of studies and reviews of psychosocial functioning done. I would contest your last sentence in this paragraph: there is a lot of research out there on ADHD and we know a lot about it already. Arguably, what we still do not know (or what is still contested) is what sort of interventions help adolescents with ADHD.

Answer: Thank you for your suggestion. This paragraph has been shortened according to your comment. Most of the studies previously cited were based on data from children and adolescents combined into one group, so it was our overstatement to discuss them. The last sentence in this paragraph was imprecise and has also been changed in the following way:

“The preliminary searches have shown that still relatively little is known about the psychosocial functioning of adolescents with ADHD.”

4.3. Objectives of the study and review questions

A. Consider moving both of these into the introduction. I was looking for this information earlier on. According to JBI these should be in the introduction of a protocol.

Answer: As explained previously (see Answer #4.2.A) these parts are the subsections of the Introduction.

B. P5,l105/6: “this study assumes taking into account..” this sentence doesn’t really make sense – I would recommend deleting it.

Answer: Thank you for the suggestion. As the objectives of the study were described in more detail, we have decided not to delete this sentence, but to change it in the following way:

“This description will contain information both on psychosocial functioning in general and in particular life environments (family, school, peer group).”

C. The objectives section still does not make very clear what the actual main aim of this study is apart from gathering information on psychosocial functioning in adolescents with ADHD. Why do you want to gather this information?

Answer: Thank you for your comment, the objectives section has been expanded and clarified. The introduced changes have emphasized that our goal is descriptive – to identify, characterise and summarise research evidence on a given topic, including the identification of research gaps. Such a goal, according to the scoping review of scoping reviews (Pham et al., 2014), is the main objective of most scoping reviews and it is sufficient for this method.

4.4. Eligibility criteria

A. Concept, p7,l158: There really needs to be some explanation of “psychosocial functioning” – this is your main concept. There needs to be a definition (see also line163 in which you say “as defined in this review” but you haven’t defined it). 

Answer: Thanks for the suggestion. The Concept subsection has been rewritten and clarified.

B. The mentioning of the three environments could be moved to the context section – this is the context in which you are looking at the concept.

Answer: After careful consideration of this comment, we have decided not to move environments issue to the Context section. According to the definitions mentioned in the Introduction, psychosocial functioning is described as the ability of an individual to function in various social roles (Priebe, 2007). Functioning in a school, family or peer group are therefore situated within the semantic limits of psychosocial functioning and – for some researchers – may be components of this concept. Our conclusion is in line with JBI's explanation of the context, which is rather related to cultural factors such as geographic location (e.g. country) or specific social or gender-based interests. In some cases, the context may also encompass details of a specific setting (e.g., healthcare system). However, none of the above factors apply to our scoping review, for which each contextual setting will be eligible for inclusion.

C. P7, l160-166: This is all not relevant to the concept. This can go into exclusion criteria.

Answer: Thank you for the comment. This paragraph was originally intended to clarify various questionable situations. However, in light of your comment, we have realised that this could lead to confusion and misunderstandings. Therefore, the paragraph has been deleted. There was also no need to move it to the Exclusion criteria as it described the specific circumstances of inclusion. Such details were unnecessary and more confusing, and did not change anything about the actual inclusion criteria. See also Answer #4.1.E.

D. P8, l175/176: why are you excluding systematic reviews and meta-analyses?

Answer: Thank you for drawing attention to this issue. Indeed, there was no justification for excluding systematic reviews and meta-analyses. This deficiency has been corrected by adding an explanation in the Search strategy and Discussion sections (as another limitation of the study).

E. P8,l188/193: for these exclusion criteria it is, again, really important that you have a solid definition of psychosocial functioning somewhere. Just presenting some examples and then writing “etc”., does not seem a full list of exclusions or inclusion. Also, if you are excluding all studies without clinical diagnosis and also those that do not fully conceptualise psychosocial functioning in a “holistic” way (however you want to define that) -my guess is you will be left with very few or no studies.

Answer: Thank you for the comment. The definition issue has been corrected (see Answer #3). The comment concerns the 4th and 5th criteria from the Exclusion criteria subsection, which have been replaced by one criterion: 

“(…) 4. no use of any of the following terms for the variables of interest: psychosocial functioning, social functioning, family functioning, school functioning, academic functioning, peer functioning, peer group functioning, or adequate grammatical forms;”

We hope that these changes have clarified this issue.

4.5. Source of evidence selection

P10 – according to JBI methodology the title/abstract screening and full text selection should be done by 2 reviewers independently.

Answer: Thank you for focusing attention on this oversight. This was noticed during the course of the project (but after the submission of the manuscript) and the procedure has been corrected. Therefore, the description of the procedure in the article has been modified accordingly.

4.6. Data analysis and presentation

P12,l 272: “Although the synthesis of the results is not the goal of the scoping review,..”: what is your goal then? You may need to re-phrase this. The goal of any review should be the synthesis of evidence in some form!

Answer: Thank you for the comment, this part of the sentence has been deleted. We understand that our sentence may not have been clear to the reader. However, it seems to us that this is a disagreement resulting from a different understanding of the word synthesis. The JBI manual explicitly states that: “formal synthesis is not normally conducted in a scoping review” (Peters et al., 2020, p.411), and further: “It is important to point out that scoping reviews do not synthesize the results/outcomes of included sources of evidence as this is more appropriately done within the conduct of a systematic review” (p.421).

REVIEWER #2:

1. The authors stated that other/similar review have been conducted “relatively long ago”. Would it be possible for the authors to explicitly state a timeframe? For example, “more than 5 years ago”.

Answer: Thank you for the comment. According to the changes suggested by Reviewer #1 (see Answer #4.2.H) this sentence has been deleted. After the reviews, we concluded that most of the studies previously cited based on data from children and adolescents combined into one group . Therefore, it was not entirely appropriate to mention them in the protocol referring only to research on the adolescent population.

2. The authors used “etc” a number of times in the manuscript. Can the authors rephrase these sentences to avoid using “etc” as this is less common for academic submissions?

Answer: Thank you for drawing attention to this issue. Sentences containing “etc” have been rephrased or deleted (due to changes resulting from other comments).

3. The authors describe the propose study as innovative. However, the argument that “ADHD is still being researched and we still know relatively little about it in youth” is a bit unclear. The authors may wish to further develop this point possibly narrowing it to focus on psychosocial difficulties in adolescents. This is important as there is a wealth of evidence on ADHD and youth as it relates to other fields like medicine and nutrition.

Answer: Than you for your comment. Indeed, this sentence was vague and has been clarified in the following way:

“The preliminary searches have shown that still relatively little is known about the psychosocial functioning of adolescents with ADHD.”

4. The authors may wish to provide a citation to support their justification for excluding information not published in peer reviewed journals. This is important as one of the advantages of the scoping review methodology is that is gives researchers the opportunities to search more broadly using grey literature sources to identify information that is usually missed in traditional systematic reviews. It is possible that grey literature searches were conducted in the preliminary searches as the authors mentioned Figshare and OSF. 

Answer: Thank you for the suggestion. After the changes made, the Discussion section cites studies in which grey literature was excluded. Balancing our desire for a broad review (seven large databases, articles since 1987, and three basic living environments) with a reasonable time frame and research project resources, we decided to include only highly reliable, well-designed studies. We are aware that this is a limitation of our study and this issue is emphasized in the Discussion section.

Indeed, grey literature was included in the preliminary search (PROSPERO, OSF and figshare.com checked). However, this only served to determine if there are ongoing or existing scoping reviews and/or systematic reviews on psychosocial functioning of adolescents with ADHD (to avoid duplication of the research).

5. The authors described piloting 10% of the studies during screening and data extraction before independently working on the remaining 90%. However, it is not clear if any verification or cross-checking or comparisons will be done to ensure further accuracy, reliability and consistency. Although not explicitly stated in the JBI guidance would the authors consider this as a limitation?

Answer: Thank you for focusing attention on this issue. According to the comment of the Reviewer #1 (see Answer #4.5.), the description of the procedure has been modified in the article. In line with the JBI methodology, the title/abstract screening and full text selection should be done by 2 (or more) reviewers independently. This oversight in the procedure was noticed during the course of the project (but after the submission of the manuscript) and the procedure has been corrected. 90% of the studies will be assessed in detail by two independent reviewers against inclusion criteria. The results will be compared and disagreements between reviewers will be resolved through discussion and consensus of the research team. The JBI methodology does not assume any verification/comparison of the data extraction process results. Therefore, we have decided not to consider this as a limitation of this scoping review. Of course, any doubts during the data extraction process will be discussed by team members.

6. Similarly, the authors may wish to acknowledge that not including the “optional consultation phase” proposed by other scoping review methodologist (Arksey and O’Malley framework) could be a limitation or a consideration for future research? Owing to the research questions proposed there might be an opportunity here for valuable input from lived experience experts.

Answer: Thank you for this valuable information on the possible benefits of consulting experienced experts. This will come in handy in our future research. However, in this scoping review, we would like to consistently apply the JBI methodology. Therefore, we did not include the proposed phase (to avoid combining methodological approaches), and we have decided not to consider this as a limitation of this scoping review. It seems to us that suggesting an “optional consultation phase” as consideration for future research will be more relevant and useful in the final scoping review report (when the results of the review are known) than in this protocol.

---

## [Decision Letter · Decision Letter 1]

23 May 2022

Psychosocial functioning of adolescents with ADHD in the family, school and peer group: A scoping review protocol

PONE-D-21-37767R1

Dear Dr.Swietek ,

We’re pleased to inform you that your manuscript has been judged scientifically suitable for publication and will be formally accepted for publication once it meets all outstanding technical requirements.

Kind regards,

Gerard Hutchinson, MD

Academic Editor

PLOS ONE

Additional Editor Comments (optional):

You should engage the suggestions to avoid repetition. 

Reviewers' comments:

Reviewer's Responses to Questions

**Comments to the Author**

1. Does the manuscript provide a valid rationale for the proposed study, with clearly identified and justified research questions?

Reviewer #1: Yes

Reviewer #2: Yes

2. Is the protocol technically sound and planned in a manner that will lead to a meaningful outcome and allow testing the stated hypotheses?

Reviewer #1: Yes

Reviewer #2: Yes

3. Is the methodology feasible and described in sufficient detail to allow the work to be replicable?

Reviewer #1: Yes

Reviewer #2: Yes

4. Have the authors described where all data underlying the findings will be made available when the study is complete?

Reviewer #1: Yes

Reviewer #2: Yes

5. Is the manuscript presented in an intelligible fashion and written in standard English?

Reviewer #1: Yes

Reviewer #2: Yes

6. Review Comments to the Author

You may also provide optional suggestions and comments to authors that they might find helpful in planning their study.

Reviewer #1: Thank you for revising the manuscript and resubmitting it. Overall, I think it is now much better, clearer and easier to follow. I got a much better understanding of what the authors are suggesting. I have a few minor suggestions to add, mainly I tend to think that the manuscript could be shortened a little by avoiding repetition here and there. Please take these as suggestions, if the editors are happy with the length this does not need to be changed on my account.

Abstract, L37: Suggest you reword the sentence to have the databases searched after the colon at the end: “The following databases will be searched for primary studies in peer-reviewed journals, written in English and published since 1987: Academic Search…”

Introduction:

I think this is much better now both in content and in wording, even though quite long (might be possible to shorten it a bit?). I would take out lines 148-153 – I don’t think you need to explain this in the introduction. You provided a good overview of what psychosocial functioning is and have explained that there are different ways to conceptualise it – I would leave this detail for the methods – but just a suggestion.

The revised section about rationale (l185-212) is also much improved. I feel this could also be shortened a bit, you have a few sentences in there like “As mentioned,..”, which indicates that there is a bit of repetition in here. Overall, it is now much clearer what gap you are trying to fill with your research and why!

Concept/Context:

I still think your life environments of family, school and peer group could be the context in which you are looking at the concept – but as you have given this some thought and have decided against this suggestion, I am sure you can make it work this way as well.

Reviewer #2: The authors have sufficiently addressed all previous comments within the text and provided appropriate justifications for methodological decisions.

7. PLOS authors have the option to publish the peer review history of their article (what does this mean?). If published, this will include your full peer review and any attached files.

Reviewer #1: No

Reviewer #2: **Yes: **Shaun Liverpool

---

## [Editor Report · Acceptance letter]

10 Jun 2022

PONE-D-21-37767R1 

Psychosocial functioning of adolescents with ADHD in the family, school and peer group: A scoping review protocol 

Dear Dr. Karteczka-Świętek:

I'm pleased to inform you that your manuscript has been deemed suitable for publication in PLOS ONE. Congratulations! Your manuscript is now with our production department. 

Kind regards, 

on behalf of

Dr. Gerard Hutchinson 

Academic Editor

PLOS ONE